# ROBUST META-LEARNING WITH NOISE VIA EIGEN-REPTILE

## ABSTRACT

Recent years have seen a surge of interest in meta-learning techniques for tackling the few-shot learning (FSL) problem. However, the meta-learner's initial model is prone to meta-overfit, as there are only a few available samples with sampling noise. Besides, when handling the data sampled with label noise for FSL, meta-learner could be extremely sensitive to label noise. To address these two challenges that FSL with sampling and label noise. In particular, we first cast the meta-overfitting problem (overfitting on sampling and label noise) as a gradient noise problem since few available samples cause meta-learner to overfit on existing examples (clean or corrupted) of an individual task at every gradient step. We present Eigen-Reptile (ER) that updates the meta-parameters with the main direction of historical task-specific parameters to alleviate gradient noise. Specifically, the main direction is computed by a special mechanism for the parameter's large size. Furthermore, to obtain a more accurate main direction for Eigen-Reptile in the presence of label noise, we propose Introspective Self-paced Learning (ISPL) that constructs a plurality of prior models to determine which sample should be abandoned. We have proved the effectiveness of Eigen-Reptile and ISPL, respectively, theoretically and experimentally. Moreover, our experiments on different tasks demonstrate that the proposed methods outperform or achieve highly competitive performance compared with the state-of-the-art methods with or without noisy labels.

## 1 INTRODUCTION

Meta-learning, also known as learning to learn, is the key for few-shot learning (FSL) (Vinyals et al., 2016; Wang et al., 2019a). One of the meta-learning methods is the gradient-based method, which usually optimizes meta-parameters as initialization that can fast adapt to new tasks with few samples. However, fewer samples mean a higher risk of meta-overfitting, as the ubiquitous sampling noise in mini-batch cannot be ignored. Moreover, existing gradient-based meta-learning methods are fragile with few samples. For instance, a popular recent method, Reptile (Nichol et al., 2018), updates the meta-parameters towards the inner loop direction, which is from the current initialization to the last task-specific parameters. Nevertheless, as shown by the bold line of *Reptile* in Figure 1, with the gradient update at the last step, the update direction of meta-parameters has a significant disturbance, as sampling noise leads the meta-parameters to overfit on the few trained samples at gradient steps. Many prior works have proposed different solutions for the meta-overfitting problem, such as using dropout (Bertinetto et al., 2018; Lee et al., 2020), and modifying the loss function (Jamal & Qi, 2019) etc., which stay at the model level. This paper casts the meta-overfitting problem as a gradient noise problem that from sampling noise while performing gradient update (Wu et al., 2019). Neelakantan et al. (2015) .etc have proved that adding additional gradient noise can improve the generalization of neural networks with large samples. However, it can be seen from *the model complexity penalty* that the generalization of the neural network will increase when the number of samples is larger. To a certain extent adding gradient noise is equivalent to increasing the sample size. As for FSL, there are only a few samples of each task. In that case, the model will not only remembers the contents that need to be identified but also overfits on the noise (Zhang et al., 2016).

High-quality manual labeling data is often time-consuming and expensive. Low-cost approaches to collect low-quality annotated data, such as from search engines, will introduce label noise. Moreover, training meta-learner requires a large number of tasks, so that it is not easy to guarantee the quality of data. Conceptually, the initialization learned by existing meta-learning algorithms can severely

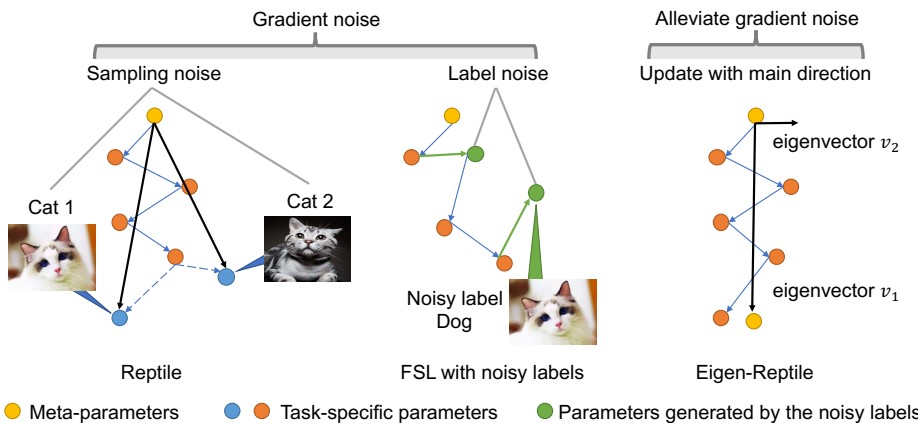

Figure 1: Inner loop steps of Reptile, Eigen-Reptile. Reptile updates meta-parameters towards the last task-specific parameters, which is biased. Eigen-Reptile considers all samples more fair. Note that the main direction is the eigenvector corresponding to the largest eigenvalue.

degrade in the presence of noisy labels. Intuitively, as shown in *FSL with noisy labels* of Figure 1, noisy labels cause a large random disturbance in the update direction. It means that label noise (Frénay & Verleysen, 2013) leads the meta-learner to overfit on wrong samples, which can be seen as further aggravating the influence of gradient noise. Furthermore, conventional algorithms about learning with noisy labels require much data for each class (Hendrycks et al., 2018; Patrini et al., 2017). Therefore, these algorithms cannot be applied to noisy FSL problem, since few available samples per class. So it is crucial to propose a method to address the problem of noisy FSL.

In this paper, we propose Eigen-Reptile (ER). In particular, as shown in Figure 1, Eigen-Reptile updates the meta-parameters with the main direction of task-specific parameters that can effectively alleviate gradient noise. Due to the large scale of neural network parameters, it is unrealistic to compute historical parameters' eigenvectors. We introduce the process of fast computing the main direction into FSL, which computes the eigenvectors of the inner loop step scale matrix instead of the parameter scale matrix. Furthermore, we propose Introspective Self-paced Learning (ISPL), which constructs multiple prior models with randomly sampling. Then prior models will discard high loss samples from the dataset. We combine Eigen-Reptile with ISPL to address the noisy FSL problem, as ISPL can improve the main direction computed with noisy labels.

Experimental results show that Eigen-Reptile significantly outperforms the baseline model by $5.35\%$ and $3.66\%$ on corrupted Mini-ImageNet of 5-way 1-shot and clean Mini-ImageNet of 5-way 5-shot, respectively. Moreover, the proposed algorithms outperform or are highly competitive with state-of-the-art methods on few-shot classification tasks.

The main contributions of this paper can be summarized as follows:

- We cast the meta-overfitting issue (overfitting on sampling and label noise) as a gradient noise issue under the meta-learning framework.
- We propose Eigen-Reptile that can alleviate gradient noise effectively. Besides, we propose ISPL, which improves the performance of Eigen-Reptile in the presence of noisy labels.
- The proposed methods outperform or achieve highly competitive performance compared with the state-of-the-art methods on few-shot classification tasks.

## 2 RELATED WORK

There are three main types of meta-learning approaches: metric-based meta-learning approaches (Ravi & Larochelle, 2016; Hochreiter et al., 2001; Andrychowicz et al., 2016; Liu et al., 2018; Santoro et al., 2016), model-based meta-learning approaches (Vinyals et al., 2016; Koch et al., 2015; Mordatch, 2018; Sung et al., 2018; Snell et al., 2017; Oreshkin et al., 2018; Shyam et al., 2017) and

gradient-based meta-learning approaches (Finn et al., 2017; Nichol et al., 2018; Jamal & Qi, 2019; Zintgraf et al., 2018; Li et al., 2017; Rajeswaran et al., 2019; Finn et al., 2018). In this paper, we focus on gradient-based meta-learning approaches which can be viewed as the bi-level loop. The goal of the outer loop is to update the meta-parameters on a variety of tasks, while task-specific parameters are learned through only a small amount of data in the inner loop. In addition, some algorithms achieve state-of-the-art results by additionally training a 64-way classification task on meta-training set (Yang et al., 2020; Hu et al., 2020), we do not compare these algorithms for fairness.

**Meta-Learning with overfitting.** Due to too few samples, meta-learner inevitably tends to overfit in FSL. Zintgraf et al. (2018) introduces additional context parameters to the model's parameters, which can prevent meta-overfitting. Furthermore, Bertinetto et al. (2018) find that regularization such as dropout can alleviate meta-overfitting in their prior work; Jamal & Qi (2019) propose a novel paradigm of Task-Agnostic Meta-Learning (TAML), which uses entropy or other approaches to minimize the inequality of initial losses beyond the classification tasks to improve the generalizability of meta-learner. All these methods stay at the model level. However, we solve the meta-overfitting problem from the gradient aspect. We propose Eigen-Reptile, which updates the meta-parameters by the main direction of task-specific parameters to alleviate meta-learner overfit on noise.

**Learning with noisy labels.** Learning with noisy labels has been a long-standing problem. There are many approaches to solve it, such as studying the denoise loss function (Hendrycks et al., 2018; Patrini et al., 2017; Jindal et al., 2016; Patrini et al., 2017; Wang et al., 2019b; Arazo et al., 2019), relabeling (Lin et al., 2014), and so on. Nevertheless, most of these methods require much data for each class. Gao et al. (2019) proposes a model for noisy few-shot relation classification but without good transferability. For noisy FSL, a gradient-based meta-learner is trained to optimize an initialization on various tasks with noisy labels. As there are few samples of each class, the traditional algorithms for noisy labels cannot be applied. When the existing gradient-based meta-learning algorithms, such as Reptile, update meta-parameters, they focus on the samples that generate the last gradient step. And these samples may be corrupted, which makes the parameters learned by meta-learner susceptible to noisy labels. For noisy FSL, we propose ISPL based on the idea of Self-paced Learning (SPL) (Kumar et al., 2010; Khan et al., 2011; Basu & Christensen, 2013; Tang et al., 2012) to learn more accurate main direction for Eigen-Reptile. ISPL constructs prior models to decide which sample should be discarded when train task-specific models, and this process can be regarded as the introspective process of meta-learner. In contrast, the model with SPL learns the samples gradually from easy to complex, and the model itself decides the order.

## 3 PRELIMINARIES

Gradient-based meta-learning aims to learn a set of initialization parameters $\phi$ that can be adapted to new tasks after a few iterations. The dataset $D$ is usually divided into the meta-training set $D_{meta-train}$ and meta-testing set $D_{meta-test}$, which are used to optimize meta-parameters and evaluate its generalization, respectively. For meta-training, we have tasks $\{\mathcal{T}_i\}_{i=1}^{B}$ drawn from task distribution $p(\mathcal{T})$, each task has its own train set $D_{train}$ and test set $D_{test}$, and the tasks in meta-testing $D_{meta-test}$ are defined in the same way. Note that there are only a small number of samples for each task in FSL. Specifically, the N-way K-shot classification task refers to K examples for each of the N classes. Generally, the number of shots used in meta-training should match the one used at meta test-time to obtain the best performance (Cao et al., 2019). In this paper, we will increase the sample size appropriately to get the main direction of individual tasks during meta-training. To minimize the test loss $L(D_{test}, \widetilde{\phi})$ of an individual task, meta-parameters need to be updated $n$ times to get good task-specific parameters $\widetilde{\phi}$. That is minimizing

$$L(D_{test}, \widetilde{\phi}) = -\frac{1}{N}\sum \mathbb{E}\left[\frac{1}{K}\sum_{(x,y)\in\mathcal{D}_{test}} \log q\left(\hat{y} = y \mid x, \phi, \widetilde{\phi}\right)\right], \quad \widetilde{\phi} = U^n(D_{train}, \phi) \quad (1)$$

where $U^n$ represents $n$ inner loop steps through gradient descent or Adam (Kingma & Ba, 2014) on batches from $D_{train}$, $q\left(\hat{y} = y \mid x, \phi, \widetilde{\phi}\right)$ is the predictive distribution. When considering updating the meta-parameters in the outer loop, different algorithms have different rules. In the case of Reptile, after $n$ inner loop steps, the meta-parameters can be updated as: $\phi \longleftarrow \phi + \beta(\widetilde{\phi} - \phi)$, where $\beta$ is a scalar stepsize hyperparameter that controls the update rate of meta-parameters.

## 4 EIGEN-REPTILE FOR CLEAN AND CORRUPTED DATA

The proposed Eigen-Reptile alleviates gradient noise to alleviate meta-learner overfitting on sampling and label noise. Furthermore, ISPL improves the performance of Eigen-Reptile in noisy FSL.

### 4.1 THE EIGEN-REPTILE ALGORITHM

To alleviate gradient noise to improve the generalizability of meta-learner, we propose Eigen-Reptile, which updates $d$ meta-parameters with the main direction of task-specific parameters. We train the task-specific model with $n$ inner loop steps that start from the meta-parameters $\phi$ with few examples. Let i-th column $\boldsymbol{W}_{:,i} \in R^{d \times 1}$ of parameter matrix $\boldsymbol{W} \in R^{d \times n}$ be the parameters after i-th gradient update, e.g., $\boldsymbol{W}_{:,i} = U^i(\phi)$. And treat $\boldsymbol{W}_{:,i}$ as a $d$-dimensional parameter point $\boldsymbol{w}_i$ in the parameter space. $\boldsymbol{e} \in R^{d \times 1}$ is a unit vector that represents the main direction of $n$ parameter points in $\boldsymbol{W}$. Intuitively, projecting all parameter points onto $\boldsymbol{e}$ should retain the most information.

We represent the parameter points by a straight line of the form $\boldsymbol{w} = \overline{\boldsymbol{w}} + l\boldsymbol{e}$, which shows that the straight line passes through the mean point $\overline{\boldsymbol{w}}$ and the signed distance of a point $\boldsymbol{w}$ from $\overline{\boldsymbol{w}}$ is $l$. Then we get the loss function $J(l_1, l_2, \cdots, l_n, \boldsymbol{e}) = \sum_{i=1}^{n} \| \overline{\boldsymbol{w}} + l_i\boldsymbol{e} - \boldsymbol{w}_i \|^2$. And determine the signed distance $l$ of each point by partially differentiating $J$ with respect to $l_i$, we get $l_i = \boldsymbol{e}^\top(\boldsymbol{w}_i - \overline{\boldsymbol{w}})$. Plugging in this expression for $l_i$ in $J$, we get

$$J(\boldsymbol{e}) = -\sum_{i=1}^{n} \boldsymbol{e}^\top(\boldsymbol{w}_i - \overline{\boldsymbol{w}})(\boldsymbol{w}_i - \overline{\boldsymbol{w}})^\top \boldsymbol{e} + \sum_{i=1}^{n} \| \boldsymbol{w}_i - \overline{\boldsymbol{w}} \|^2 = -\boldsymbol{e}^\top \boldsymbol{S} \boldsymbol{e} + \sum_{i=1}^{n} \| \boldsymbol{w}_i - \overline{\boldsymbol{w}} \|^2 \quad (2)$$

where $\boldsymbol{S} = \sum_{i=1}^{n}(\boldsymbol{w}_i - \overline{\boldsymbol{w}})(\boldsymbol{w}_i - \overline{\boldsymbol{w}})^\top$ is a scatter matrix. According to Eq.2, minimizing $J$ is equivalent to maximizing: $\boldsymbol{e}^\top \boldsymbol{S} \boldsymbol{e}$. Note that $\boldsymbol{e}$ needs to be roughly consistent with the gradient update direction $\overline{\boldsymbol{V}}$ in the process of learning task-specific parameters. Use Lagrange multiplier method as

$$\max \boldsymbol{e}^\top \boldsymbol{S} \boldsymbol{e} \quad \text{s.t.} \begin{cases} \overline{\boldsymbol{V}} \boldsymbol{e} > 0 \\ \boldsymbol{e}^\top \boldsymbol{e} = 1 \end{cases}, \text{where} \quad \overline{\boldsymbol{V}} = \frac{1}{\lfloor n/2 \rfloor} \sum_{i=1}^{\lfloor n/2 \rfloor} \boldsymbol{w}_{n-i+1} - \boldsymbol{w}_i \quad (3)$$

We get the objective function

$$g(\mu, \boldsymbol{e}, \lambda, \eta) = \boldsymbol{e}^\top \boldsymbol{S} \boldsymbol{e} - \lambda(\boldsymbol{e}^\top \boldsymbol{e} - 1) + \mu(-\overline{\boldsymbol{V}}\boldsymbol{e} + \eta^2), \quad \text{where} \quad \lambda \neq 0, \mu \geq 0 \quad (4)$$

then partially differentiating $g$ in Eq.4 with respect to $\mu, \boldsymbol{e}, \lambda, \eta$,

$$\begin{cases} -\overline{\boldsymbol{V}}\boldsymbol{e} + \eta^2 = 0 \\ 2\boldsymbol{S}\boldsymbol{e} - 2\lambda\boldsymbol{e} - \mu\overline{\boldsymbol{V}} = 0 \\ \boldsymbol{e}^\top \boldsymbol{e} - 1 = 0 \\ 2\mu\eta = 0 \end{cases} \quad (5)$$

According to Eq.5, if $\eta = 0$, then $\overline{\boldsymbol{V}}$ and $\boldsymbol{e}$ are orthogonal, which obviously does not meet our expectations. So we get $\eta \neq 0$, and $\mu = 0$. Then $\boldsymbol{S}\boldsymbol{e} = \lambda\boldsymbol{e}$. We can see $\boldsymbol{e}$ is the eigenvector of $\boldsymbol{S}$ corresponding to the largest eigenvalue $\lambda$, as we need the main direction. It should be noted that even if Eq.3 is not directly related to the $\boldsymbol{e}$, in ER, Eq.3 must be retained because it determines the update direction of the outer-loop. Otherwise, the algorithm will not converge.

A concerned question about $\boldsymbol{S}\boldsymbol{e} = \lambda\boldsymbol{e}$ is that the scatter matrix $\boldsymbol{S} \in R^{d \times d}$ grows quadratically with the number of parameters $d$. As the high dimensionality of parameters typically used in neural networks, computing eigenvalue and eigenvector of $\boldsymbol{S}$ could come at a prohibitive cost (the worst-case complexity is $\mathcal{O}(\mathrm{d}^3)$ ). Centralize $\boldsymbol{W}$ by subtracting the mean $\overline{\boldsymbol{w}}$, and scatter matrix $\boldsymbol{S} = \boldsymbol{W}\boldsymbol{W}^\top$. To avoid calculating the eigenvector of $\boldsymbol{S}$ directly, we focus on $\boldsymbol{W}^\top \boldsymbol{W}$. As $\boldsymbol{W}^\top \boldsymbol{W}\widehat{\boldsymbol{e}} = \widehat{\lambda}\widehat{\boldsymbol{e}}$. Multiply both sides of the equation with $\boldsymbol{W}$,

$$\boldsymbol{W}\boldsymbol{W}^\top \underbrace{\boldsymbol{W}\widehat{\boldsymbol{e}}}_{\boldsymbol{e}} = \underbrace{\widehat{\lambda}}_{\lambda}\underbrace{\boldsymbol{W}\widehat{\boldsymbol{e}}}_{\boldsymbol{e}} \quad (6)$$

It can be found from Eq.6 that $\boldsymbol{W}^\top \boldsymbol{W} \in R^{n \times n}$ and $\boldsymbol{W}\boldsymbol{W}^\top \in R^{d \times d}$ have the same eigenvalue, $\lambda = \widehat{\lambda}$. Furthermore, we get the eigenvector of $\boldsymbol{W}\boldsymbol{W}^\top$ as $\boldsymbol{e} = \boldsymbol{W}\widehat{\boldsymbol{e}}$. The main advantage of Eq.6 is

that the intermediate matrix $\boldsymbol{W}^\top \boldsymbol{W}$ now grows quadratically with the inner loop steps. As we are interested in FSL, $n$ is very small. It will be much easier to compute the eigenvector $\widehat{\boldsymbol{e}}$ of $\boldsymbol{W}^\top \boldsymbol{W}$, $\mathcal{O}\left(n^3\right)$, which can be ignored (detailed analysis refer to **Appendix** B). Then we get the eigenvector $\boldsymbol{e}$ of $\boldsymbol{W}\boldsymbol{W}^\top$ based on $\widehat{\boldsymbol{e}}$. Moreover, we project parameter update vectors $\boldsymbol{w}_{i+1} - \boldsymbol{w}_i, i = 1, 2, \cdots, n-1$ on $\boldsymbol{e}$ to get the corresponding update stepsize $\nu$, so meta-parameters $\phi$ can be updated as

$$\phi \longleftarrow \phi + \beta\nu\boldsymbol{e} \tag{7}$$

where $\beta$ is a scalar stepsize hyperparameter that controls the update rate of meta-parameters. The Eigen-Reptile algorithm is summarized in Algorithm 1 of **Appendix A**. To illustrate the validity of Eigen-Reptile theoretically, we present Theorem 1 as follow:

**Theorem 1** *Assume that the gradient noise variable $x$ follows Gaussian distribution (Hu et al., 2017; Jastrzębski et al., 2017; Mandt et al., 2016), $x \sim \mathrm{N}\left(0, \sigma^2\right)$. Furthermore, $x$ and neural network parameter variable are assumed to be uncorrelated. The observed covariance matrix $\boldsymbol{C}$ equals noiseless covariance matrix $\boldsymbol{C}_t$ plus gradient noise covariance matrix $\boldsymbol{C}_x$. Then, we get*

$$\boldsymbol{C} = \boldsymbol{C}_t + \boldsymbol{C}_x = \boldsymbol{P}_t(\Lambda_t + \Lambda_x)\boldsymbol{P}_t^\top = \boldsymbol{P}_t(\Lambda_t + \sigma^2\boldsymbol{I})\boldsymbol{P}_t^\top = \boldsymbol{P}_t\Lambda\boldsymbol{P}_t^\top = \boldsymbol{P}\Lambda\boldsymbol{P}^\top \tag{8}$$

*where $\boldsymbol{P}_t$ and $\boldsymbol{P}$ are the orthonormal eigenvector matrices of $\boldsymbol{C}_t$ and $\boldsymbol{C}$ respectively, $\Lambda_t$ and $\Lambda$ are the corresponding diagonal eigenvalue matrices, and $\boldsymbol{I}$ is an identity matrix. It can be seen from Eq.8 that $\boldsymbol{C}$ and $\boldsymbol{C}_t$ has the same eigenvectors. We defer the proof to the **Appendix** C.*

Theorem 1 shows that **eigenvectors are not affected by gradient noise.** So Eigen-Reptile can find a more generalizable starting point for new tasks without overfitting on noise.

## 4.2 THE INTROSPECTIVE SELF-PACED LEARNING

Self-paced learning (SPL) learns the samples from low losses to high losses, which is proven beneficial in achieving a better generalization result (Khan et al., 2011; Basu & Christensen, 2013; Tang et al., 2012). Besides, the losses of samples are determined by the model itself. Nevertheless, in meta-learning setting, meta-learner is trained on various tasks, and initial model may have lower losses for trained classes and higher losses for unseen classes or noisy samples. For this reason, we cannot train the task-specific model by the same way as SPL to solve noisy FSL problem. In this paper, we use multiple prior models to vote on samples and decide which should be abandoned. As shown in Figure 2, even though the two categories of yellow and green show an excellent distribution that can be well separated, some samples are marked wrong. To address this noisy label problem, we build three prior models. Specially, we randomly sample three times,

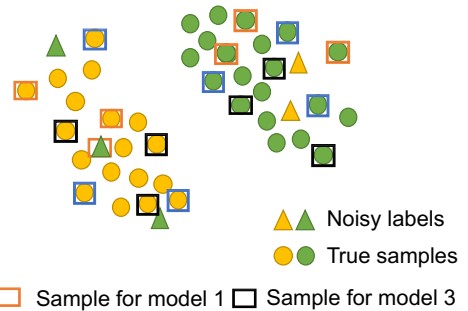

Figure 2: Randomly sample examples to build different prior models.

and model 1 is trained with a corrupted label. Due to different samples learned by prior models, building multiple models to vote on the data will obtain more accurate losses. Such a learning process is similar to human introspection, and we call it Introspective Self-paced Learning (ISPL). Moreover, samples with losses above a certain threshold will be discarded. Furthermore, we imitate SPL to add the hidden variable $v = 0$ or $1$ that is decided by $Q$ prior models before the loss of each sample to control whether the sample should be abandoned. So we get the task-specific loss as

$$L_{ISPL}\left(\boldsymbol{\phi}, \boldsymbol{v}\right) = \sum_{i=1}^{h} v_i L\left(x_i, y_i, \boldsymbol{\phi}\right), \text{where} \quad v_i = \arg\min_{\mathbf{v}} \frac{v_i}{Q} \sum_{j=1}^{Q} L_j\left(x_i, y_i, \boldsymbol{\phi}_j\right) - \gamma v_i \tag{9}$$

where $h$ is the number of samples from dataset $D_{train}$, $\gamma$ is the sample selection parameter, which gradually decreases, parameter of model $j$ is $\boldsymbol{\phi}_j = U^n(D_j, \boldsymbol{\phi}), D_j \in D_{train}$. Note that we update the meta-parameters with the model trained on $h$ samples from $D_{train}$. The ISPL is summarized in Algorithm 2 of **Appendix A**. Intuitively, it is difficult to say whether discarding high-loss samples containing correct samples and wrong samples will improve the accuracy of eigenvector, so we will use Theorem 2 to prove the effectiveness of ISPL.

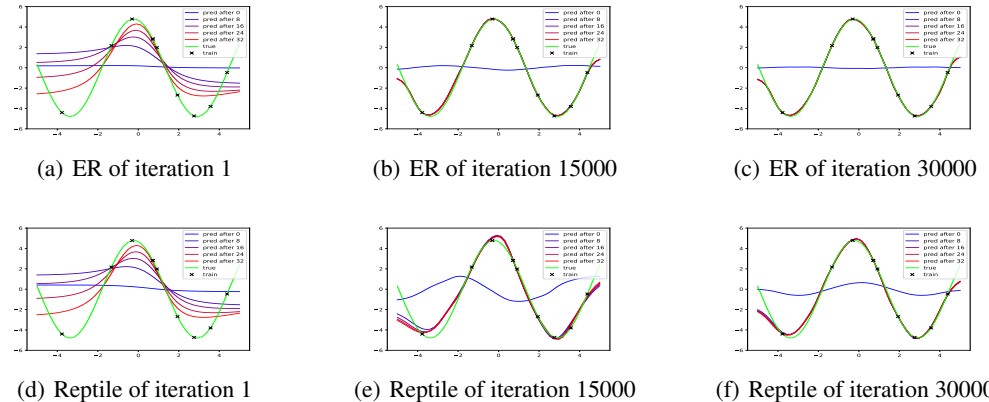

(a) ER of iteration 1      (b) ER of iteration 15000      (c) ER of iteration 30000

(d) Reptile of iteration 1      (e) Reptile of iteration 15000      (f) Reptile of iteration 30000

Figure 3: Eigen-Reptile and Reptile training process on the regression toy test. (a), (b), (c) and (d), (e), (f) show that after the gradient update 0, 8, 16, 24, 32 times based on initialization learned by Eigen-Reptile and Reptile respectively.

**Theorem 2** *Let $W_o$ be the parameter matrix generated by the corrupted samples. Compute the eigenvalues and eigenvectors of the expected observed parameter matrix*

$$\frac{1}{\lambda}\mathbb{E}(C_{tr})e = P_o(I - \frac{\Lambda_o}{\lambda})P_o^\top e \approx P_o(I - \frac{\lambda_o}{\lambda}I)P_o^\top e > P_o(I - \frac{\lambda_o - \xi}{\lambda - \xi}I)P_o^\top e \quad (10)$$

*where $C_{tr}$ is the covariance matrix generated by true samples, $\lambda$ is the observed largest eigenvalue, $\lambda_o$ is the largest eigenvalue in the corrupted diagonal eigenvalue matrix $\Lambda_o$. According to Eq.21, if $\lambda_o/\lambda$ is smaller, the observed eigenvector $e$ is more accurate. Assume that the discarded high loss samples have the same contributions $\xi$ to $\lambda$ and $\lambda_o$, representing the observed and corrupted main directional variance, respectively. Note that these two kinds of data have the same effect on the gradient updating of the model, so this assumption is relatively reasonable. Furthermore, it is easy to find that $(\lambda_o - \xi)/(\lambda - \xi)$ is smaller than $\lambda_o/\lambda$.*

Theorem 2 shows that **discard the high loss samples can help improve the accuracy of the observed eigenvector of parameter matrix learned with corrupted labels.** So ISPL can improve the performance of Eigen-Reptile, as it discards high loss samples in $D_{train}$.

## 5 EXPERIMENTAL RESULTS AND DISCUSSION

In our experiments, we aim to (1) evaluate the effectiveness of Eigen-Reptile to alleviate gradient noise (sampling and label noise), (2) determine whether Eigen-Reptile can alleviate gradient noise in a realistic problem, (3) evaluate the improvement of ISPL to Eigen-Reptile in the presence of noisy labels, (4) validate theoretical analysis through numerical simulations. The code and data for the proposed model are provided for research purposes [1].

### 5.1 META-LEARNING WITH NOISE ON REGRESSION

In this experiment, we evaluate Eigen-Reptile by the 1D sine wave $K$-shot regression problem (Nichol et al., 2018). Each task is defined by a sine curve $y(x) = Asin(x + b)$, where the amplitude $A \sim U([0.1, 5.0])$ and phase $b \sim U([0, 2\pi])$. The amplitude $A$ and phase $b$ are varied between tasks. The goal of each task is to fit a sine curve with the data points sampled from the corresponding $y(x)$. We calculate loss in $\ell_2$ using 50 equally-spaced points from the whole interval $[-5.0, 5.0]$ for each task. The loss is

$$\int_{-5.0}^{5.0} \| y(x) - \widehat{y}(x) \|^2 \, dx \quad (11)$$

---

[1]Code is included in the supplemental material. Code will be released upon the paper acceptance.

Table 1: Few Shot Classification on Mini-Imagenet N-way K-shot accuracy. The $\pm$ shows $95\%$ confidence interval over tasks.

| Algorithm | 5-way 1-shot | 5-way 5-shot |
|---|---|---|
| MAML (Finn et al., 2017) | $48.70 \pm 1.84\%$ | $63.11 \pm 0.92\%$ |
| Relation Network (Sung et al., 2018) | $50.44 \pm 0.82\%$ | $65.32 \pm 0.70\%$ |
| CAML (512) (Zintgraf et al., 2018) | $51.82 \pm 0.65\%$ | $65.85 \pm 0.55\%$ |
| TAML (VL + Meta-SGD) (Jamal & Qi, 2019) | $51.77 \pm 1.86\%$ | $65.60 \pm 0.93\%$ |
| Meta-dropout(Lee et al., 2019) | $51.93 \pm 0.67\%$ | $67.42 \pm 0.52\%$ |
| Warp-MAML (Flennerhag et al., 2019) | $52.30 \pm 0.8\%$ | $68.4 \pm 0.6\%$ |
| MC (128) (Park & Oliva, 2019) | $\mathbf{54.08 \pm 0.93}\%$ | $67.99 \pm 0.73\%$ |
| ARML (Yao et al., 2020) | $50.42 \pm 1.73\%$ | $-$ |
| ModGrad (64) (Simon et al., 2020) | $\mathbf{53.20 \pm 0.86}\%$ | $69.17 \pm 0.69$ |
| Reptile (32)(Nichol et al., 2018) | $49.97 \pm 0.32\%$ | $65.99 \pm 0.58\%$ |
| Eign-Reptile (32) | $51.80 \pm 0.9\%$ | $68.10 \pm 0.50\%$ |
| Eign-Reptile (64) | $\mathbf{53.25 \pm 0.45}\%$ | $\mathbf{69.85 \pm 0.85}\%$ |

where $\widehat{y}(x)$ is the predicted function that start from the initialization learned by meta-learner.

The $K$-shot regression task fits a selected sine curve through $K$ points, here $K = 10$. For the regressor, we use a small neural network, which is the same as Nichol et al. (2018), except that the activation functions are Tanh. Specifically, the small network includes an input layer of size 1, followed by two hidden layers of size 64, and then an output layer of size 1. In this part, we mainly compare Reptile and Eigen-Reptile. Both meta-learners use the same regressor and are trained for 30000 iterations with inner loop steps 5, batch size 10, and a fixed inner loop learning rate $\alpha = 0.02$.

We report the results of Reptile and Eigen-Reptile in Figure 3. It can be seen that the curve fitted by Eigen-Reptile is closer to the true green curve, which shows that Eigen-Reptile performs better. According to Jamal & Qi (2019), the initial model that has a larger entropy before adapting to new tasks would better alleviate meta-overfitting. As shown in Figure 3, from 1 to 30000 iterations, Eigen-Reptile is more generalizable than Reptile as the initial blue line of Eigen-Reptile is closer to a straight line, which shows that the initialization learned by Eigen-Reptile is less affected by gradient noise. Furthermore, Figure 4 shows that Eigen-Reptile converges faster and gets a lower loss than Reptile.

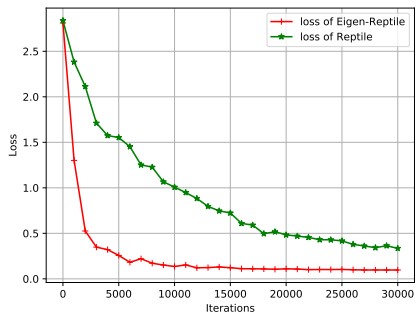

Figure 4: Loss of the 10-shot regression.

## 5.2 META-LEARNING IN REALISTIC PROBLEM

We evaluate our method on a popular few-shot classification dataset: Mini-ImageNet (Vinyals et al., 2016). The Mini-Imagenet dataset contains 100 classes, each with 600 images. We follow Ravi & Larochelle (2016) to divide the dataset into three disjoint subsets: meta-training set, meta-validation set, and meta-testing set with 64 classes, 16 classes, and 20 classes, respectively. And we follow the few-shot learning protocols from prior work (Vinyals et al., 2016), except that the number of the meta-training shot is 15, which is still much smaller than the number of samples required by traditional tasks. Moreover, we run our algorithm on the dataset for the different number of shots and compare our results to the state-of-the-art results. What needs to be reminded is that approaches that use deeper, residual networks can achieve higher accuracies (Gidaris & Komodakis, 2018), so for a fair comparison, we only compare algorithms that use convolutional networks as Reptile does. Specifically, our model follows Nichol et al. (2018), which has 4 modules with a $3 \times 3$ convolutions and 64 filters, $2 \times 2$ max-pooling etc.. The images are downsampled to $84 \times 84$, and the loss function is the cross-entropy error. We use the Adam optimizer with $\beta_1 = 0$ in the inner loop. Our model

Table 2: Average test accuracy of 5-way 1-shot on the Mini-Imagenet with symmetric label noise.

| Algorithm | p=0.0 | p=0.1 | p=0.2 | p=0.5 |
|---|---|---|---|---|
| Reptile (Nichol et al., 2018) | 47.64% | 46.08% | 43.49% | 23.33% |
| Reptile+ISPL | 47.23% | 46.50% | 43.70% | 21.83% |
| Eign-Reptile | **47.87**% | 47.18% | 45.01% | 27.23% |
| Eigen-Reptile+ISPL | 47.26% | **47.20**% | **45.49**% | **28.68**% |

is trained for 100000 iterations with a fixed inner loop learning rate 0.0005, and 7 inner-loop steps. Regarding some hyperparameters analysis, defer to Appendix E.

The results of Eigen-Reptile and other meta-learning approaches are summarized in Table 1. The proposed Eigen-Reptile (64 filters) outperforms and achieves highly competitive performance compared with other algorithms for 5-shot and 1-shot classification problems, respectively. More specifically, for 1-shot, the result of MC (128 filters) with a higher capacity network is better than that of Eigen-Reptile. However, as a second-order optimization algorithm, the computational cost of MC will be much higher than Eigen-Reptile. And obviously, the result of Eigen-Reptile is much better than Reptile for each task. Compared with Reptile, Eigen-Reptile uses the main direction to update the meta-parameters to alleviate the meta-overfitting caused by gradient noise. More importantly, Eigen-Reptile outperforms the state-of-the-art meta-overfitting preventing method Meta-dropout (Lee et al., 2019), which is based on regularization. This result shows that the effectiveness of addressing the meta-overfitting problem from the perspective of alleviating gradient noise.

### 5.3 META-LEARNING WITH LABEL NOISE

We conduct the 5-way 1-shot experiment with noisy labels generated by corrupting the original labels of Mini-Imagenet. More specifically, in this section, we only focus on symmetric label noise, as correct labels are flipped to other labels with equal probability, i.e., in the case of symmetric noise of ratio $p$, a sample retains the correct label with probability $1 - p$. It becomes some other label with probability $p/(N - 1)$. An example of symmetric noise is shown in Figure 6. Furthermore, the asymmetric label noise experiment is conducted in Appendix F. Note that we only introduce noise in the train set during meta-training, where the meta-training shot is 30. Moreover, all meta-learners with 32 filters are trained for 10000 iterations, with a learning rate of 0.001 in the inner loop. The sample selection parameter $\gamma = 10$ that decreases by 0.6 every 1000 iterations. The other settings of this experiment are the same as in section 5.2.

As shown in Table 3, with the increase of the ratio $p$, the performance of Reptile decreases rapidly. When $p = 0.5$, the initialization point learned by Reptile can hardly meet the requirements of quickly adapting to new tasks with few samples. On the other hand, Eigen-Reptile is less affected by noisy labels than Reptile, especially when the noise ratio is high, i.e., $p = 0.5$. The experimental results also verify the effectiveness of ISPL, as Eigen-Reptile+ISPL achieves better results than Eigen-Reptile when $p \neq 0$. It also can be seen that ISPL plays a more significant role when $p$ is higher. However, when $p = 0$, ISPL harms Eigen-Reptile, as ISPL only discards correct samples. In addition, ISPL does not significantly improve Reptile, especially when $p = 0.5$. This is because too many high-loss samples are removed, causing Reptile to fail to converge quickly with the same number of iterations. These experimental results show that Eigen-Reptile and ISPL can effectively alleviate the gradient noise problem caused by noisy labels, thereby alleviating the meta-overfitting on corrupted samples.

### 6 CONCLUSION

In this paper, we cast the meta-overfitting problem (overfitting on sampling and label noise) as a gradient noise problem. Then, we propose a gradient-based meta-learning algorithm Eigen-Reptile. It updates the meta-parameters through the main direction, which has been proven by theory and experiments that it can alleviate the gradient noise effectively. Furthermore, to get closer to real-world situations, we introduce noisy labels into the meta-training dataset, and the proposed ISPL constructs prior models to select samples for Eigen-Reptile to get more accurate main direction.

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

## A PSEUDO-CODE

---

**Algorithm 1** Eigen-Reptile

---

**Require:** Distribution over tasks $P(\mathcal{T})$, outer step size $\beta$
 1: Initialize meta-parameters $\boldsymbol{\phi}$
 2: **while** not converged **do**
 3:     $\boldsymbol{W} = [\,], \nu = 0$
 4:     Sample batch of tasks $\{\mathcal{T}_i\}_{i=1}^{B} \sim P(\mathcal{T})$
 5:     **for** each task $\mathcal{T}_i$ **do**
 6:         $\boldsymbol{\phi}_i = \boldsymbol{\phi}$
 7:         Sample train set $D_{train}$ of $\mathcal{T}_i$
 8:         **for** $j = 1, 2, 3, ..., n$ **do**
 9:             $\boldsymbol{\phi}_i^j = U^j(D_{train}, \boldsymbol{\phi}_i)$
10:             $\boldsymbol{W}$ appends $\boldsymbol{w}_j = flatten(\boldsymbol{\phi}_i^j), \quad \boldsymbol{w}_j \in R^{d \times 1}$
11:         **end for**
12:         Mean centering, $\boldsymbol{W} = \boldsymbol{W} - \overline{\boldsymbol{w}}, \quad \overline{\boldsymbol{w}} \in R^{d \times 1}$
13:         Compute eigenvalue matrix $\widehat{\boldsymbol{\Lambda}}$ and eigenvector matrix $\widehat{\boldsymbol{P}}$ of scatter matrix $\boldsymbol{W}^\top \boldsymbol{W}$
14:         Eigenvalues $\lambda_1 > \lambda_2 > \cdots > \lambda_n$ in $\widehat{\boldsymbol{\Lambda}}$
15:         Compute eigenvector matrix of $\boldsymbol{W}\boldsymbol{W}^\top$, $\boldsymbol{P} = \boldsymbol{W}\widehat{\boldsymbol{P}}$
16:         Let the eigenvector corresponding to $\lambda_1$ be a unit vector $\|\boldsymbol{e}_i^1\|_2^2 = 1$
17:         **for** $j = 1, 2, 3, ..., n - 1$ **do**
18:             $\nu = \nu + (\boldsymbol{W}_{:,j+1} - \boldsymbol{W}_{:,j})\boldsymbol{e}_i^1$
19:         **end for**
20:         $\boldsymbol{e}_i^1 = \frac{\lambda_1}{\sum_{m=1}^{n} \lambda_m} \times \boldsymbol{e}_i^1$
21:         Calculate the approximate direction of task-specific gradient update $\overline{\boldsymbol{V}}$:
22:         $\overline{\boldsymbol{V}} = \frac{1}{\lfloor n/2 \rfloor} \sum_{i=1}^{\lfloor n/2 \rfloor} \boldsymbol{W}_{:,n-i+1} - \boldsymbol{W}_{:,i}$
23:         **if** $\boldsymbol{e}_i^1 \cdot \overline{\boldsymbol{V}} < 0$ **then**
24:             $\boldsymbol{e}_i^1 = -\boldsymbol{e}_i^1$
25:         **end if**
26:     **end for**
27:     Average the main directions to get $\tilde{\boldsymbol{e}} = (1/B) \sum_{i=1}^{B} \boldsymbol{e}_i^1$
28:     Update meta-parameters $\boldsymbol{\phi} \longleftarrow \boldsymbol{\phi} + \beta \times \nu/B \times \tilde{\boldsymbol{e}}$
29: **end while**

---

**Algorithm 2** Introspective Self-paced Learning

---

**Require:** Dataset $D_{train}$, initialization $\boldsymbol{\phi}$, batch size $b$, selection parameter $\gamma$, attenuation coefficient $\mu$, the number of prior models $Q$
 1: Initialize network parameters $\boldsymbol{\phi}^* = \boldsymbol{\phi}$ for a sampled task
 2: **for** $j = 1, 2, 3, \cdots, Q$ **do**
 3:     Sample examples $D_j$ from $D_{train}$ for training $model_j$, $\boldsymbol{\phi}_j = U^m(D_j, \boldsymbol{\phi}^*)$
 4: **end for**
 5: Train task-specific parameters:
 6: **for** $i = 1, 2, 3, \cdots, n$ **do**
 7:     Compute hidden variable vector $\boldsymbol{v}$:
 8:     $v = \arg\min_{\mathbf{v}} v_q \sum_{q=1}^{b} L_q - \gamma \sum_{q=1}^{b} v_q$, where $L_q = \frac{1}{Q} \sum_{j=1}^{Q} L_j(x_q, y_q, \boldsymbol{\phi}_j)$
 9:     Update task-specific parameters $\boldsymbol{\phi}^*$:
10:     $\boldsymbol{\phi}^* = \arg\min_{\boldsymbol{\phi}^*} L_{ISPL}(\boldsymbol{\phi}^*, v)$
11:     $\gamma = \gamma - \mu$
12: **end for**

---

## B ALGORITHM COMPLEXITY ANALYSIS

As for Eigen-Reptile, the cost of single gradient descent in the inner-loop is $\mathcal{O}(d)$, where $d$ is the number of network parameters. The cost of the covariance matrix computations is $\mathcal{O}(n^2 d)$,

where $n$ is the number of inner-loop. Moreover, the worst-case complexity of computing eigenvalue decomposition is $\mathcal{O}(n^3)$. Finally, the computational complexity of restoring eigenvector is $\mathcal{O}(nd)$. We set the maximal number of outer-loop to $T$. Hence the overall time complexity is $\mathcal{O}(T(n^2d + n^3 + nd + nd))$. As in FSL, $n$ is usually less than 10 (for this paper $n = 7$ ), so the overall time complexity is $\mathcal{O}(Td)$. As for Reptile, the computational complexity is also $\mathcal{O}(Td)$, which means that the time complexity of both Reptile and Eigen-Reptile is much lower than the second-order optimization algorithms.

As for spatial complexity, Eigen-Reptile needs to store a $d \times n$ matrix and a $n \times n$ matrix. The overall space complexity is $\mathcal{O}(d)$, while the spatial complexity of Reptile is $\mathcal{O}(d)$, too.

It can be seen that, compared to Reptile, Eigen-Reptile is the same in spatial complexity and time complexity. Still, its accuracy is much higher than that of Reptile.

## C  THEOREM 1

Gradient update always with gradient noise inserted at every iteration, which caused Reptile, MAML, etc. cannot find accurate directions to update meta-parameters. In this section, we will prove that Eigen-Reptile can alleviate gradient noise.

**Theorem 3** *Assume that the gradient noise variable $x$ follows Gaussian distribution (Hu et al., 2017; Jastrzębski et al., 2017; Mandt et al., 2016), $x \sim \mathrm{N}\left(0, \sigma^2\right)$. Furthermore, $x$ and neural network parameter variable are assumed to be uncorrelated. The observed covariance matrix $C$ equals noiseless covariance matrix $C_t$ plus gradient noise covariance matrix $C_x$. Then, we get*

$$C = C_t + C_x = P_t(\Lambda_t + \Lambda_x)P_t^\top = P_t(\Lambda_t + \sigma^2 I)P_t^\top = P\Lambda P^\top = P_t\Lambda P_t^\top \qquad (12)$$

*where $P_t$ and $P$ are the orthonormal eigenvector matrices of $C_t$ and $C$ respectively, $\Lambda_t$ and $\Lambda$ are the corresponding diagonal eigenvalue matrices, and $I$ is an identity matrix. It can be seen from Eq.12 that $C$ and $C_t$ has the same eigenvectors.*

**Proof C.1** *In the following proof, we assume that the probability density function of gradient noise variable $x$ follows Gaussian distribution, $x \sim \mathrm{N}\left(0, \sigma^2\right)$. Treat the parameters in the neural network as variables, and the parameters obtained by each gradient update as samples. Furthermore, gradient noise and neural network parameters are assumed to be uncorrelated.*

*For observed parameter matrix $W \in R^{d \times n}$, there are $n$ samples, let $W_{i,:} \in R^{1 \times n}$ be the observed values of the $i$-th variable $W_i$, and $W = [W_{1,:}^\top, \cdots, W_{i,:}^\top, \cdots, W_{d,:}^\top]^\top$. Similarly, we denote the noiseless parameter matrix by $W^t = [(W_{1,:}^t)^\top, \cdots, (W_{i,:}^t)^\top, \cdots, (W_{d,:}^t)^\top]^\top$, and*

$$W = W^t + X \qquad (13)$$

*Where $X = [X_{1,:}^\top, \cdots, X_{i,:}^\top, \cdots, X_{d,:}^\top]^\top$ is the dataset of noise variables. Then, centralize each variable by*

$$\overline{W}_k = W_k - \frac{1}{n}\sum_{i=1}^{n} W_{k,:}(i) \qquad (14)$$

*So we get $\overline{W} = [\overline{W}_1^\top, \cdots, \overline{W}_d^\top]^\top$. Suppose $W^t$ is also centralized by the same way and get $\overline{W^t} = [\overline{W^t}_1^\top, \cdots, \overline{W^t}_d^\top]^\top$. Then, we have:*

$$\overline{W} = \overline{W^t} + X \qquad (15)$$

*Computing the covariance matrix of $\overline{W}$:*

$$\begin{aligned} C &= \frac{1}{n}\overline{W}\overline{W}^\top \\ &= \frac{1}{n}(\overline{W^t} + X)(\overline{W^t}^\top + X^\top) \\ &= \frac{1}{n}(\overline{W^t}\overline{W^t}^\top + \overline{W^t}X^\top + X\overline{W^t}^\top + XX^\top) \end{aligned} \qquad (16)$$

Since $\overline{\boldsymbol{W}^t}$ and $\boldsymbol{X}$ are uncorrelated, $\overline{\boldsymbol{W}^t}\boldsymbol{X}^\top$ and $\boldsymbol{X}\overline{\boldsymbol{W}^t}^\top$ are approximately zero matrices. Thus:

$$\boldsymbol{C} \approx \frac{1}{n}(\overline{\boldsymbol{W}^t\boldsymbol{W}^t}^\top + \boldsymbol{X}\boldsymbol{X}^\top) = \boldsymbol{C}_t + \boldsymbol{C}_x \tag{17}$$

The component $\boldsymbol{C}_x(i,j)$ is the correlation between $\boldsymbol{X}_i$ and $\boldsymbol{X}_j$ which corresponds to the i-th and j-th rows of $\boldsymbol{X}$. As the two noise variables are not related to each other, if $i \neq j$, then $\boldsymbol{C}_x(i,j) = 0$. So $\boldsymbol{C}_x \in R^{d\times d}$ is a diagonal matrix with diagonal elements $\sigma^2$. Decompose $\boldsymbol{C}_t$ as:

$$\boldsymbol{C}_t = \boldsymbol{P}_t \boldsymbol{\Lambda}_t \boldsymbol{P}_t^\top \tag{18}$$

where $\boldsymbol{P}_t$ is the noiseless orthonormal eigenvector matrix and $\boldsymbol{\Lambda}_t$ is the noiseless diagonal eigenvalue matrix, then

$$\boldsymbol{C}_x = \boldsymbol{\Lambda}_x \boldsymbol{P}_t \boldsymbol{P}_t^\top = \boldsymbol{P}_t \boldsymbol{\Lambda}_x \boldsymbol{P}_t^\top = \boldsymbol{P}_t \boldsymbol{C}_x \boldsymbol{P}_t^\top \tag{19}$$

where $\boldsymbol{\Lambda}_x = \sigma^2 \boldsymbol{I}$, and $\boldsymbol{I}$ is the identity matrix. Thus,

$$\begin{aligned}
\boldsymbol{C} &= \boldsymbol{C}_t + \boldsymbol{C}_x \\
&= \boldsymbol{P}_t \boldsymbol{\Lambda}_t \boldsymbol{P}_t^\top + \boldsymbol{P}_t \boldsymbol{\Lambda}_x \boldsymbol{P}_t^\top. \\
&= \boldsymbol{P}_t (\boldsymbol{\Lambda}_t + \boldsymbol{\Lambda}_x) \boldsymbol{P}_t^\top. \\
&= \boldsymbol{P}_t \boldsymbol{\Lambda} \boldsymbol{P}_t^\top
\end{aligned} \tag{20}$$

where $\boldsymbol{\Lambda} = \boldsymbol{\Lambda}_t + \boldsymbol{\Lambda}_x$. It can be seen from Eq.20 that $\boldsymbol{C}$ and $\boldsymbol{C}_t$ has the same eigenvector matrix. In other words, **eigenvector is not affected by gradient noise.**

## D   THEOREM 2

In this section, we will prove that discarding high loss samples will result in a more accurate main direction in noisy FSL.

**Theorem 4** *Let $\boldsymbol{W}_o$ be the parameter matrix generated by the corrupted samples. Compute the eigenvalues and eigenvectors of the expected observed parameter matrix*

$$\frac{1}{\lambda}\mathbb{E}(\boldsymbol{C}_{tr})\boldsymbol{e} = \boldsymbol{P}_o(\boldsymbol{I} - \frac{\boldsymbol{\Lambda}_o}{\lambda})\boldsymbol{P}_o^\top \boldsymbol{e} \approx \boldsymbol{P}_o(\boldsymbol{I} - \frac{\lambda_o}{\lambda}\boldsymbol{I})\boldsymbol{P}_o^\top \boldsymbol{e} > \boldsymbol{P}_o(\boldsymbol{I} - \frac{\lambda_o - \xi}{\lambda - \xi}\boldsymbol{I})\boldsymbol{P}_o^\top \boldsymbol{e} \tag{21}$$

*where $\boldsymbol{C}_{tr}$ is the covariance matrix generated by true samples, $\lambda$ is the observed largest eigenvalue, $\lambda_o$ is the largest eigenvalue in the corrupted diagonal eigenvalue matrix $\boldsymbol{\Lambda}_o$. According to Eq.21, if $\lambda_o/\lambda$ is smaller, the observed eigenvector $\boldsymbol{e}$ is more accurate. Assume that the discarded high loss samples have the same contributions $\xi$ to $\lambda$ and $\lambda_o$, representing the observed and corrupted main directional variance, respectively. Note that these two kinds of data have the same effect on the gradient updating of the model, so this assumption is relatively reasonable. Furthermore, it is easy to find that $(\lambda_o - \xi)/(\lambda - \xi)$ is smaller than $\lambda_o/\lambda$.*

**Proof D.1** *Here, we use $\boldsymbol{w}$ to represent the parameter point obtained after a gradient update. For convenience, let $\boldsymbol{w}$ be generated by a single sample, $\boldsymbol{w} \in R^{d\times 1}$. Then the parameter matrix can be obtained,*

$$\boldsymbol{W} = \begin{bmatrix} \boldsymbol{w}_1^{tr} & \boldsymbol{w}_2^{tr} & \cdots & \boldsymbol{w}_1^o & \cdots & \boldsymbol{w}_m^o & \cdots & \boldsymbol{w}_n^{tr} \end{bmatrix} \tag{22}$$

*where $\boldsymbol{w}^o$ represents the parameters generated by the corrupted sample, and $\boldsymbol{w}^{tr}$ represents the parameters generated by the true sample. Furthermore, there are $n$ parameter points generated by $n$ samples. Moreover, there are $m$ corrupted parameter points generated by $m$ corrupted samples. Mean centering $\boldsymbol{W}$, and show the observed covariance matrix $\boldsymbol{C}$ as*

$$\begin{aligned}
\boldsymbol{C} &= \frac{1}{n}\boldsymbol{W}\boldsymbol{W}^\top \\
&= \frac{1}{n}\begin{bmatrix} \boldsymbol{w}_1^{tr} & \boldsymbol{w}_2^{tr} & \cdots & \boldsymbol{w}_n^{tr} \end{bmatrix} \begin{bmatrix} (\boldsymbol{w}_1^{tr})^\top \\ (\boldsymbol{w}_2^{tr})^\top \\ \vdots \\ (\boldsymbol{w}_n^{tr})^\top \end{bmatrix} \\
&= \frac{1}{n}(\boldsymbol{w}_1^{tr}(\boldsymbol{w}_1^{tr})^\top + \cdots + \boldsymbol{w}_1^o(\boldsymbol{w}_1^o)^\top + \cdots + \boldsymbol{w}_m^o(\boldsymbol{w}_m^o)^\top + \cdots + \boldsymbol{w}_n^{tr}(\boldsymbol{w}_n^{tr})^\top)
\end{aligned} \tag{23}$$

*It can be seen from the decomposition of $C$ that the required eigenvector is related to the parameters obtained from the true samples and the parameters obtained from the noisy samples. For a single parameter point*

$$\boldsymbol{w}\boldsymbol{w}^\top = \begin{bmatrix} a_1 \\ \vdots \\ a_i \\ \vdots \\ a_d \end{bmatrix} \begin{bmatrix} a_1 & \cdots & a_i & \cdots & a_d \end{bmatrix} \tag{24}$$

$$= \begin{bmatrix} a_1^2 & a_1 a_2 & \cdots & a_1 a_d \\ a_2 a_1 & a_2^2 & \cdots & a_2 a_d \\ \vdots & \vdots & \ddots & \vdots \\ a_d a_1 & a_d a_2 & \cdots & a_d^2 \end{bmatrix}$$

*As we discard all high loss samples that make the model parameters change significantly, and the randomly generated noisy labels may cause the gradient to move in any direction, we assume that the variance of corrupted parameter point variables is $\delta$. Compute the expectations of all variables in the corrupted parameter point*

$$\mathbb{E}(\boldsymbol{w}\boldsymbol{w}^\top) = \begin{bmatrix} \delta_1^2 + \mathbb{E}(a_1)^2 & \mathbb{E}(a_1 a_2) & \cdots & \mathbb{E}(a_1 a_d) \\ \mathbb{E}(a_2 a_1) & \delta_2^2 + \mathbb{E}(a_2)^2 & \cdots & \mathbb{E}(a_2 a_d) \\ \vdots & \vdots & \ddots & \vdots \\ \mathbb{E}(a_d a_1) & \mathbb{E}(a_d a_2) & \cdots & \delta_d^2 + \mathbb{E}(a_d)^2 \end{bmatrix} = \Omega \tag{25}$$

*Let the sum of all corrupted $\frac{1}{n}\mathbb{E}(\boldsymbol{w}\boldsymbol{w}^\top)$ be $\boldsymbol{\Omega}_o$, then*

$$\boldsymbol{\Omega}_o = \frac{1}{n} \begin{bmatrix} m\delta^2 + \sum_{j=1}^m \mathbb{E}(a_{j1})^2 & \cdots & \sum_{j=1}^m \mathbb{E}(a_{j1} a_{jd}) \\ \sum_{j=1}^m \mathbb{E}(a_{j2} a_{j1}) & \cdots & \sum_{j=1}^m \mathbb{E}(a_{j2} a_{jd}) \\ \vdots & \ddots & \vdots \\ \sum_{j=1}^m \mathbb{E}(a_{jd} a_{j1}) & \cdots & m\delta^2 + \sum_{j=1}^m \mathbb{E}(a_{jd})^2 \end{bmatrix} \tag{26}$$

*And let the sum of all true $\frac{1}{n}\boldsymbol{w}\boldsymbol{w}^\top$ be $\boldsymbol{C}_{tr}$. So the expectation of $\boldsymbol{C}$ can be written as,*

$$\mathbb{E}(\boldsymbol{C}) = \mathbb{E}(\boldsymbol{C}_{tr}) + \boldsymbol{\Omega}_o \tag{27}$$

*Treat eigenvector and eigenvalue as definite values, we get*

$$(\boldsymbol{\Omega}_o + \mathbb{E}(\boldsymbol{C}_{tr}))\boldsymbol{e} = \lambda \boldsymbol{e} \tag{28}$$

*where $\boldsymbol{e}$ is the observed eigenvector, $\lambda$ is the corresponding eigenvalue. Divide both sides of the equation by $\lambda$.*

$$\frac{1}{\lambda}\mathbb{E}(\boldsymbol{C}_{tr})\boldsymbol{e} = (\boldsymbol{I} - \frac{1}{\lambda}\boldsymbol{\Omega}_o)\boldsymbol{e}$$

$$= \boldsymbol{P}_o(\boldsymbol{I} - \frac{1}{\lambda}\boldsymbol{\Lambda}_o)\boldsymbol{P}_o^\top \boldsymbol{e} \tag{29}$$

$$\approx \boldsymbol{P}_o(\boldsymbol{I} - \frac{\lambda_o}{\lambda}\boldsymbol{I})\boldsymbol{P}_o^\top \boldsymbol{e}$$

*where $\lambda_o$ is the largest eigenvalue in the corrupted diagonal eigenvalue matrix $\boldsymbol{\Lambda}_o$, $\boldsymbol{P}_o$ is the orthonormal eigenvector matrix of $\boldsymbol{\Omega}_o$. According to Eq.29, if $\lambda_o/\lambda$ is smaller, $\boldsymbol{e}$ is more accurate. Discard some samples with the largest losses, which may contain true samples and noisy samples. Assume that the discarded high loss samples have the same contributions $\xi$ to $\lambda$ and $\lambda_o$, as these two kinds of data have the same effect on the gradient updating of the model. Compare the ratio of eigenvalues before and after discarding, get*

$$\underbrace{\frac{\lambda_o}{\lambda}}_{before} - \underbrace{\frac{\lambda_o - \xi}{\lambda - \xi}}_{after}$$

$$= \frac{\xi(\lambda - \lambda_o)}{\lambda(\lambda - \xi)} > 0 \tag{30}$$

*Obviously, $\lambda > \lambda_o$, and if we don't discard all samples, then $\lambda > \xi$. So Eq.30$> 0$, which means discarding high loss samples could reduce $\lambda_o/\lambda$. **Therefore, discarding high loss samples can improve the accuracy of eigenvector in the presence of noisy labels.***

*For further analysis, we assume that any two variables are independently and identically distributed, the expectation of variable $a$, $\mathbb{E}(a) = \epsilon$. Thus,*

$$\frac{1}{\lambda}\mathbf{\Omega}_o = \frac{p}{\lambda}\begin{bmatrix} \delta^2 + \epsilon^2 & \cdots & \epsilon^2 \\ \epsilon^2 & \cdots & \epsilon^2 \\ \vdots & \ddots & \vdots \\ \epsilon^2 & \cdots & \delta^2 + \epsilon^2 \end{bmatrix} \tag{31}$$

*where $p$ is the proportion of noisy labels, $np = m$. As can be seen from Eq.31, if $p\epsilon^2/\lambda \approx 0$, then $\mathbf{\Omega}_o/\lambda$ is a diagonal matrix. According to proof. C.1, the observed eigenvector $\mathbf{e}$ is unaffected by noisy labels with the corresponding eigenvalue $\frac{p(\delta^2+\epsilon^2)}{\lambda}$.*

## E  HYPERPARAMETERS OF EIGEN-REPTILE

In this section, we follows Lee et al. (2019); Cao et al. (2019) to vary the number of inner-loops and the number of corresponding training shots to show the robustness of Eigen-Reptile. Besides, other hyperparameters are the same as section 5.2. As shown in Figure 5, after the number of inner-loops $i$ reaches 7, the test accuracy tends to be stable, which shows that changing the number of inner-loops within a certain range has little effect on Eigen-Reptile. That is, Eigen-Reptile is robust to this hyperparameter. As for train shot, to make the trained task-specific parameters as unbiased as possible, we specify train shot roughly satisfies $\lceil \frac{i \times batch\_size}{N} \rceil + 1$, where $N$ is the number of classes. So when $i = 7$, the number of train shots is 15. It is important to note that in our experiments, Reptile uses the original implementation hyperparameters, the number of inner-loops is 8, the number of train shots is 15 and the corresponding accuracy is 65.99%.

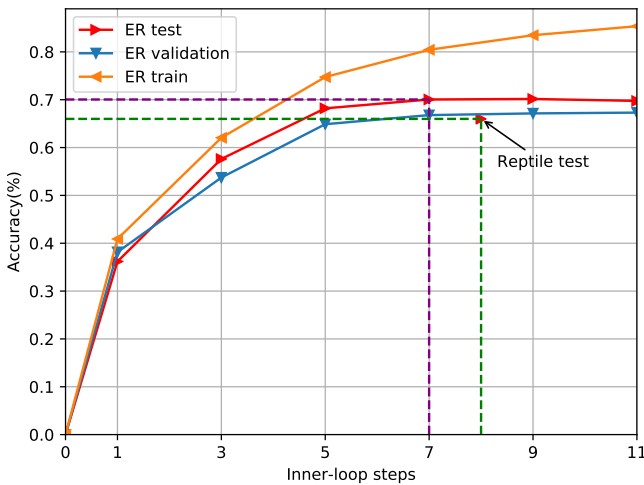

Figure 5:  The number of inner-loop and accuracy of 5-way 5-shot task on Mini-Imagenet.

## F  THE ASYMMETRIC LABEL NOISE EXPERIMENT

Since asymmetric noise is as common as symmetric noise, we focus on asymmetric noise in this section. As illustrated in *asymmetric noise* of Figure 6, we randomly flip the labels of one class to the labels of another class without duplication in the meta-training dataset (64 classes).

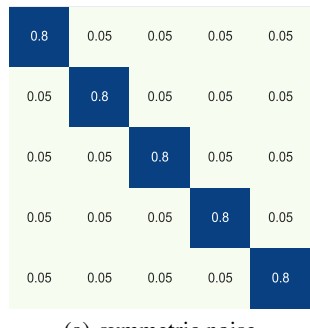 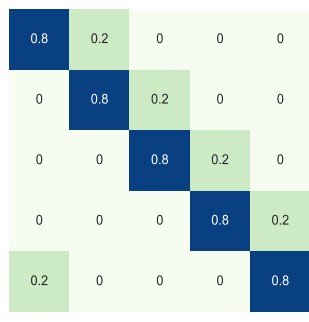

|             (a) symmetric noise             |             (b) asymmetric noise             |

Figure 6: Examples of noise transition matrix (taking 5 classes and noise ratio 0.2 as an example).

We compare our algorithms with Reptile in table 3. We observe that meta-learning algorithms with asymmetric noise have closer results compared to symmetric noise. As meta-learner is trained on tasks with the same noise transition matrix, which allows the meta-learner to learn more useful information, the results are higher than that with symmetric noise. Similar to the symmetric noise results, Eigen-Reptile outperforms Reptile in all tasks, and ISPL plays a more significant role when $p$ is higher. The experimental results also show that when the number of iterations is constant, ISPL does not significantly improve or even degrades the results of Reptile. On the contrary, ISPL can provide Eigen-Reptile with a more accurate main direction in difficult tasks, e.g., $p = 0.2, 0.5$.

Table 3: Average test accuracy of 5-way 1-shot on the Mini-Imagenet with asymmetric label noise.

| Algorithm | p=0.1 | p=0.2 | p=0.5 |
|---|---|---|---|
| Reptile (Nichol et al., 2018) | 47.30% | 45.51% | 42.03% |
| Reptile+ISPL | 47.00% | 45.42% | 41.09% |
| Eign-Reptile | **47.42**% | 46.50% | 42.29% |
| Eigen-Reptile+ISPL | 47.24% | **46.83**% | **43.71**% |

## G  META-LEARNING ON CIFAR-FS

Bertinetto et al. (2018) propose CIFAR-FS (CIFAR100 few-shots), which is randomly sampled from CIFAR-100 (Krizhevsky et al., 2009), containing images of size $32 \times 32$. The settings of Eigen-Reptile in this experiment are the same as in 5.2. Moreover, we do not compare algorithms with additional tricks, such as higher way (Bertinetto et al., 2018; Snell et al., 2017). It can be seen from Table 4 that on CIFAR-FS, the performance of Eigen-Reptile is still far better than Reptile without any parameter adjustment.

Table 4:  Few Shot Classification on CIFAR-FS N-way K-shot accuracy. The $\pm$ shows 95% confidence interval over tasks.

| Algorithm | 5-way 1-shot | 5-way 5-shot |
|---|---|---|
| MAML (Finn et al., 2017) | $58.90 \pm 1.90\%$ | $71.50 \pm 1.00\%$ |
| PROTO NET(Snell et al., 2017) | $55.50 \pm 0.70\%$ | $72.00 \pm 0.60\%$ |
| GNN(Satorras & Estrach, 2018) | $61.90\%$ | $75.30\%$ |
| Embedded Class Models(Ravichandran et al., 2019) | $55.14 \pm 0.48\%$ | $71.66 \pm 0.39\%$ |
| Reptile (Nichol et al., 2018) | $58.30 \pm 1.20\%$ | $75.45 \pm 0.55\%$ |
| Eign-Reptile (64) | $61.90 \pm 1.40\%$ | $\mathbf{78.30 \pm 0.50}\%$ |
| Eign-Reptile (128) | $\mathbf{62.30 \pm 1.40}\%$ | $\mathbf{78.55 \pm 0.45}\%$ |

## H    NEURAL NETWORK ARCHITECTURES

This section will show the performance of Eigen-Reptile on Mini-Imagenet when using a larger network as CAML, etc. Note that we only compare our algorithm with meta-learning algorithms based on the gradient in this section. As shown in Table 5, when Eigen-Reptile uses a larger convolutional neural network (CNN), higher accuracy can be obtained, which shows that Eigen-Reptile benefits from increasing model expressiveness.

Table 5:   Few Shot Classification on Mini-Imagenet N-way K-shot accuracy.  The $\pm$ shows $95\%$ confidence interval over tasks.

| Algorithm | Backbone | 5-way 1-shot | 5-way 5-shot |
|---|---|---|---|
| MAML (Finn et al., 2017) | Conv-4-64 | $48.7 \pm 1.8\%$ | $63.1 \pm 0.9\%$ |
| Meta-SGD (Li et al., 2017) | Conv-4-64 | $50.47 \pm 1.87\%.$ | $64.03 \pm 0.94\%$ |
| CAML(Zintgraf et al., 2018) | Conv-4-32 | $47.24 \pm 0.65\%$ | $59.05 \pm 0.54\%$ |
| CAML(Zintgraf et al., 2018) | Conv-4-512 | $51.82 \pm 0.65\%$ | $65.85 \pm 0.55\%$ |
| iMAML (Rajeswaran et al., 2019) | Conv-4-64 | $49.30 \pm 1.88\%$ | $-$ |
| MC (128) (Park & Oliva, 2019) | Conv-4-128 | $\mathbf{54.08 \pm 0.93}\%$ | $67.99 \pm 0.73\%$ |
| HSML(Yao et al., 2019) | Conv-4-32 | $50.38 \pm 1.85\%$ | $-$ |
| Reptile(Nichol et al., 2018) | Conv-4-32 | $49.97 \pm 0.32\%$ | $65.99 \pm 0.58\%$ |
| Eign-Reptile | Conv-4-32 | $51.80 \pm 0.90\%$ | $68.10 \pm 0.50\%$ |
| Eign-Reptile | Conv-4-64 | $\mathbf{53.25 \pm 0.45}\%$ | $\mathbf{69.65 \pm 0.85}\%$ |

