# OpenReview forum: "Robust Meta-learning with Noise via Eigen-Reptile"
_ICLR.cc/2021/Conference — Reject_

### Official Review · AnonReviewer4 · 2020-10-24
**The manuscript needs a major revision. The experiments and analysis are poor.**

**Rating:** 5
**Confidence:** 5

**Review:**

The paper presents a method about robust meta-learning by analyzing eigenvalues and eigenvectors. The ultimate goal is to learn the new task and denoise the gradients when updating the model parameters for the specific task.  There are two aspects in the proposed approach to alleviate gradient noise: 1). Parameters updates with eigen vectors. 2). introspective self-paced learning to correct the loss of noisy samples.


Strengths:
- The problem of meta-learning with noisy labels is introduced in this work with some experiments to show the robustness of the proposed approach.
- The analysis with various noise percentages shows that meta-learning suffers from noisy labels.

Weaknesses:
- The manuscript is very hard to follow especially some important information is located in Appendix. For instance, in the main paper it is not very clear how the algorithm works and implemented, this important information is provided in Appendix (Algorithm 1 and 2).
- It is not very clear how the inner-update is performed because U(.) can be any meta-learning algorithm but the work only focuses on Reptile. The inner-loop should be explicitly defined.
- This work has no analysis about the limit of noise that is acceptable to the algorithm (especially related to the eigenvalues).
- The assumption of Theorem 1 is strong that the gradient contains Gaussian noise from the noisy labels. Is there any empirical evidence/previous work providing the fact or the proof?
- The experiment settings are very artificial and the problem is not well-motivated. How is it possible to have a lot of noisy labels when only a few data is provided? Is there any valid reason or practical example to justify the setup in this work?
- The literature review of this work is very poor:
1. Meta-learning with differentiable closed-form solvers is proposed by Bertinetto et al. [1]. Is there any reason to cite the technical report (reimplementation) by Devos et al.?
2. Gradient noise in meta-learning has been investigated by Simon et al. [2] but it is not discussed and compared in the manuscript.
3. A recent method called Meta-Curvature [3] which is a closely correlated topic to the proposed method is not discussed and compared.

- The experiments are not convincing to show the superiority of the proposed method:
1. The dataset is limited to one dataset for classification tasks. This paper only provides the mini-ImageNet dataset.
2. A rigorous experiment is needed by varying the parameters (e.g. the number of shot, the number of inner-loop, and the number of way) to show the robustness are not shown. This is important because the setup is new but the experiments do not show various conditions.
3. There is no comparison with the existing methods for combating noisy labels. Some approaches [4, 5, 6] can be adopted to the loss function for inner-loop.


There are some questions that need to be addressed:
1. In Eq. 3, is it possible that \bar{V}.e is zero? Because the inner-loop can reach a stationary point.
2. Is ISPL applicable to any learning condition? What is the special thing of this loss function in meta-learning?


[1] L. Bertinetto, J. F. Henriques,  P. Torr, and A. Vedaldi, "Meta-learning with differentiable closed-form solvers," ICLR, 2018.

[2] C. Simon, P. Koniusz, R. Nock, M. Harandi, "On modulating the gradient for meta-learning," ECCV, 2020.

[3] E. Park and J.B. Olivia, "Meta-curvature," Neurips, 2019.

[4] E. Arazo, D. Ortego, P. Albert, N. E. O'Connor, K. McGuiness,"Unsupervised Label Noise Modeling and Loss Correction," ICML, 2019.

[5] Y.Wang, X. Ma, Z. Chen, Y. Luo, J. Yi, J. Bailey, "Symmetric Cross Entropy for Robust Learning with Noisy Labels,"ICCV, 2019.

[6] G. Patrini, A. Rozza, A. K. Menon, R. Nock, L. Qu,"Making Deep Neural Networks Robust to Label Noise:
a Loss Correction Approach," CVPR, 2017.



--------------------------------------------------------------------------------------------------------------------------------------------------

I want to thank the authors for the revision and the rebuttal. Even though the proposed method is promising for meta-learning with noisy gradients, the setup is not backed up with strong arguments/facts. Moreover, I do not fully agree to the statements in the response. For instance, these arguments are not fully observable neither by experiments nor theories:

1. " Specifically, the meta-learner is prone to meta-overfit, as there are only a few available samples with sampling noise, even data is clean.". There is no proof that the meta-learner is prone to meta-overfit. The 'meta-overfit' term is not defined well. What I understand is that the learned meta-parameter is prone to overfitting in this context.
2. "Due to the small amounts of samples, FSL is more easily affected by data noise, especially considering that human annotators are likely to make mistakes as training meta-learner requires a large number of classes. Besides, we are not the first to propose noisy FSL, [10] propose hybrid attention-based prototypical networks for the noisy few-shot relation classification task, as human annotators are also likely to make mistakes in language tasks.". A common sense is that a few data is more managable and less vulnerable to mislabeling.
3. "Our noisy labels experiments aim to verify the robustness of Eigen-Reptile to noisy labels. Denoise is not the focus of our research. At the same time, ISPL is not a denoise algorithm in the traditional sense.". The statement is vague and not aligned to the introduction, abstract, and title of this work.

The theoretical part is mostly trivial as mentioned by R2, and the contribution (Algorithm) is hidden in Appendices. I acknowledge the author's response by increasing the score but I think the paper needs to be further revised.

I suggest to build strong background of this topic (meta-learning + noise/label noise) and relevant experiments to show the effectiveness of the proposed approach. Furthermore, some theoretical analysis about the limit to the noise can be verified for the future direction. For experiments with noisy gradients, I suggest to compare with the baseline on meta-learning + noise, e.g. the work by Simon et al. [2]. If the focus was noisy labels then it would be better to compare with other methods for noisy labels (not necessarily in the meta-learning setup).

---

> ### Author Response · Authors · 2020-11-18
> **Reply(Part1)**
>
> We want to thank the reviewer for your thorough reading and valuable comments! However, there are some points of misunderstanding that we address in this rebuttal.
>
> In this paper, we cast the meta-overfitting issue that overfitting on sampling noise (for clean data) and label noise (for corrupted data) as a gradient noise issue, then propose Eigen-Reptile that can alleviate gradient noise effectively. Besides, we propose ISPL, which improves the performance of Eigen-Reptile in the presence of noisy labels. We have proved the effectiveness of Eigen-Reptile and ISPL, respectively, theoretically and experimentally.
>
> Q1.The manuscript is very hard to follow especially some important information is located in Appendix. For instance, in the main paper it is not very clear how the algorithm works and implemented, this important information is provided in Appendix (Algorithm 1 and 2).
>
> Thanks for your suggestions. Although we think Alg. 1 and Alg. 2 are good summaries of Section4.1 and Section4.2, they are not critical to understand our proposed methods. This is why we put them into the appendix. When we have more space in the final version, we can move Algorithm 1 and 2 in the main paper.
>
> Q2. It is not very clear how the inner-update is performed because U(.) can be any meta-learning algorithm but the work only focuses on Reptile. The inner-loop should be explicitly defined.
>
> Here we are imitating the representation of the inner-loop operation in the Reptile. U(.) corresponds to performing gradient descent or Adam on batches of data. Any algorithm can update the inner-loop by these operations.
>
> Q3.This work has no analysis about the limit of noise that is acceptable to the algorithm (especially related to the eigenvalues).
>
> It can be seen from Table 2 that when the noise ratio reaches 50%, Reptile is already close to random, and the results of ER and ER+ISPL are relatively much better. It can be considered that 50% of the noise is still acceptable for ER and ER+ISPL. That is, the eigenvector is still meaningful. It should be noted that the corruption of the correct labels causes noisy labels. In traditional denoise algorithms, the noise ratio usually does not exceed 50%, so we think that the upper limit of noise ratio as 50% is sufficient. It does not make much sense to study a higher noise ratio even if the algorithm is acceptable. Besides, noisy FSL's experiment is only used to prove the robustness of the ER and ISPL. Noisy labels are not the focus of our research.
>
> Q4.The assumption of Theorem 1 is strong that the gradient contains Gaussian noise from the noisy labels. Is there any empirical evidence/previous work providing the fact or the proof?
>
> It must be reminded that the gradient noise mainly from sampling noise in Theorem 1. Specifically, the meta-learner is prone to meta-overfit, as there are only a few available samples with sampling noise, even data is clean.
>
> The assumption is reasonable, as the gradient noise studied in many previous works [7,8,9] is also assumed to be Gaussian noise.
>
> [7] Mandt S, Hoffman M, Blei D. A variational analysis of stochastic gradient algorithms[C]//International conference on machine learning. 2016: 354-363.
>
> [8] Jastrzębski S, Kenton Z, Arpit D, et al. Three factors influencing minima in sgd[J]. arXiv preprint arXiv:1711.04623, 2017.
>
> [9] Hu W, Li C J, Li L, et al. On the diffusion approximation of nonconvex stochastic gradient descent[J]. Annals of Mathematical Sciences and Applications, 2019, 4(1): 3-32.
>
> Q5.The experiment settings are very artificial and the problem is not well-motivated. How is it possible to have a lot of noisy labels when only a few data is provided? Is there any valid reason or practical example to justify the setup in this work?
>
> Due to the small amounts of samples, FSL is more easily affected by data noise, especially considering that human annotators are likely to make mistakes as training meta-learner requires a large number of classes. Besides, we are not the first to propose noisy FSL, [10] propose hybrid attention-based prototypical networks for the noisy few-shot relation classification task, as human annotators are also likely to make mistakes in language tasks.
>
> [10] Tianyu Gao, Xu Han, Zhiyuan Liu, and Maosong Sun. Hybrid attention-based prototypical networks for noisy few-shot relation classification. In Proceedings of the AAAI Conference on Artificial Intelligence, volume 33, pp. 6407–6414, 2019.
>
> Q6a.The literature review of this work is very poor: Meta-learning with differentiable closed-form solvers is proposed by Bertinetto et al. [1]. Is there any reason to cite the technical report (reimplementation) by Devos et al.?
>
> Thank you for your reminder. We have revised this quote in the new version.

---

> > ### Author Response · Authors · 2020-11-18
> > **Reply(Part2)**
> >
> > Q6b.Gradient noise in meta-learning has been investigated by Simon et al. [2] but it is not discussed and compared in the manuscript.
> >
> > Thank you for your reminder. We have already discussed and compared [2] in the paper.
> >
> > Q6c. A recent method called Meta-Curvature [3] which is a closely correlated topic to the proposed method is not discussed and compared.
> >
> > Thank you for your reminder. We have already discussed and compared [3] in our paper.
> >
> > Q7a. The experiments are not convincing to show the superiority of the proposed method:
> > (a)The dataset is limited to one dataset for classification tasks. This paper only provides the mini-ImageNet dataset.
> >
> > We chose mini-ImageNet because it is the most widely used in the gradient-based meta-learning algorithm. We apply the ER to CIFAR-FS, and the results are reported as follows and in Appendix G. The settings of Eigen-Reptile in this experiment are the same as in section 5.2(mini-ImageNet experiment).
> > Moreover, we do not compare algorithms with additional tricks, such as higher way [1,15]. It can be seen from Table that on CIFAR-FS, the performance of Eigen-Reptile is still far better than Reptile without any parameter adjustment.
> >
> > Method   |   5-way 1-shot  |.  5-way 5-shot
> >
> > MAML                                 |$58.90\pm 1.90\%$       | $71.50 \pm 1.00 \% $
> >
> > PROTO                                |$55.50 \pm 0.70\%$      | $72.00 \pm 0.60\%$
> >
> > Embedded Class Models |$55.14 \pm 0.48\%$      | $71.66 \pm 0.39\%$
> >
> > Reptile                                |$58.30 \pm 1.20 \%$     |$75.45\pm 0.55\%$
> >
> > Eign-Reptile                       |$62.30\pm1.40\%$          |$78.55\pm 0.45\%$
> >
> >
> > Q7b. A rigorous experiment is needed by varying the parameters (e.g. the number of shot, the number of inner-loop, and the number of way) to show the robustness are not shown. This is important because the setup is new but the experiments do not show various conditions.
> >
> > Our setup follows [11,12,13,14]. In the new version, Appendix E, we vary the number of inner-loops and the corresponding training shots to show the robustness of Eigen-Reptile.
> > Inner-loops 1,3,5,7,9,11 correspond to the results of an experiment: 0.36182, 0.57616, 0.68198, 0.7005, 0.70162, 0.6975. After the number of inner-loops $i$ reaches 7, the test accuracy tends to be stable, which shows that changing the number of inner-loops within a certain range has little effect on Eigen-Reptile. That is, Eigen-Reptile is robust to this hyperparameter. As for train shot, to make the trained task-specific parameters as unbiased as possible (avoid using one sample too many times to reduce the risk of overfitting), we specify train shot roughly satisfies $\lceil \frac{i\times batchsize}{N} \rceil+1$, where $N$ is the number of classes. So when $i=7$, the number of train shots is 15.
> >
> > As for the number of ways, it is also not a new problem. [15] proposes higher way that trains the model with 20- way on the miniImageNet 5-way classification, which can effectively improve the training effect.
> >
> > [11] Tianshi Cao, Marc Law, and Sanja Fidler. A theoretical analysis of the number of shots in few-shot learning. arXiv preprint arXiv:1909.11722, 2019.
> >
> > [12] https://github.com/openai/supervised-reptile
> >
> > [13] Lee K, Maji S, Ravichandran A, et al. Meta-learning with differentiable convex
> > optimization[C]//Proceedings of the IEEE Conference on Computer Vision and Pattern Recognition. 2019: 10657-10665.
> >
> > [14] Bertinetto L, Henriques J F, Torr P H S, et al. Meta-learning with differentiable closed-form solvers[J]. arXiv preprint arXiv:1805.08136, 2018.
> >
> > [15] Jake Snell, Kevin Swersky, and Richard Zemel. Prototypical networks for few-shot learning. In NIPS, 2017.
> >
> > Q7c. There is no comparison with the existing methods for combating noisy labels. Some approaches [4, 5, 6] can be adopted to the loss function for inner-loop.
> >
> > Our noisy labels experiments aim to verify the robustness of Eigen-Reptile to noisy labels. Denoise is not the focus of our research. At the same time, ISPL is not a denoise algorithm in the traditional sense. It discards both correct and corrupted samples to provide Eigen-Reptile with more accurate eigenvectors. The application of these denoising algorithms may improve the accuracy of the model itself. Still, they can also be used for other meta-learning algorithms and cannot be targeted to enhance the robustness of Eigen-Reptile like ISPL. We cited and discussed the difference between ISPL and these algorithms in the new version.

---

> > > ### Author Response · Authors · 2020-11-20
> > > **Reply(Part3)**
> > >
> > > Q8. In Eq. 3, is it possible that \bar{V}.e is zero? Because the inner-loop can reach a stationary point.
> > >
> > > If the inner-loop reach a stationary point, $\bar{V}=[0…0]$, $\bar{V}$.$e=0$, so it is possible. However, it should be reminded that due to the high dimension of the parameter space, it is almost impossible to reach the stationary point regardless of whether the inner loop uses SGD or Adam [16].
> > >
> > > [16] Sagun L, Bottou L, LeCun Y. Eigenvalues of the hessian in deep learning: Singularity and beyond[J], 2016.
> > >
> > > Q9. Is ISPL applicable to any learning condition? What is the special thing of this loss function in meta-learning?
> > >
> > > ISPL does not work in other learning methods. It is designed for getting more accurate eigenvectors for ER. In Table2, when $p=0$, ISPL degrades Eigen-Reptile and Reptile's accuracy, as it only discards correct samples. Simultaneously, within a certain number of iterations, better results cannot be achieved by Reptile+ISPL, even p=0.5, because too many high-loss samples are removed. As stated in Theorem 2, ISPL will discard correct samples and corrupted samples simultaneously to improve the accuracy of the observed eigenvectors so that ISPL can effectively enhance the robustness of Eigen-Reptile. Therefore, ISPL is not suitable for most learning conditions.

---

### Official Review · AnonReviewer3 · 2020-10-28
**This paper presents a reptile-based meta-learning algorithm called Eigen-Reptile for few shot learning with sampling and label nosing. When Eigen-Reptile updates meta-parameters, it leverages not only the gradient direction of different task, but also the direction of eigenvector related to parameters matrix. Besides, authors propose Introspective Self-paced Learning (ISPL) for label noise problem.**

**Rating:** 4
**Confidence:** 3

**Review:**

This paper presents a reptile-based meta-learning algorithm called Eigen-Reptile for few shot learning with sampling and label nosing. When Eigen-Reptile updates meta-parameters, it leverages not only the gradient direction of different task, but also the direction of eigenvector related to parameters matrix. Besides, authors propose Introspective Self-paced Learning (ISPL) for label noise problem. N-way-K-shot experiments on Mini-Imagenet demonstrate the effectiveness of Eigen-Reptile and 5-way-1shot with symmetric label noise experiments on Mini-Imagenet show that ISPL could alleviate the noising problem in some degree.

Strength:
1. The idea of Eigen-Reptile to alleviate gradient noise by eigenvector of parameters-related matrix is interesting and authors prove the effectiveness of the idea in theory.
2. A clever, avoiding high time complexity way to obtain the eigenvector of parameters-related matrix
3. By idea of self-paced learning with prior model to solve noisy few shot problem is reasonable and ingenious with theoretical proof.

Weakness:
1. The writing is confusing and not clear enough. For example, Para. 4.1, line 6. What is the specific meaning of “main direction of n points”, and what is suitable mathematical expression of “the unit vector e” ?
2. In Mini-Imagenet N-way K-shot experiments, authors didn’t show specific numbers of filter of the most important comparison object,  Reptile, and the final experiments results are not particularly outstanding compared with recent papers, like DPGN[1], SIB[2].
3. For label noise experiments, it is hard to say the ISPL is indeed effective as results showed in line 1,2,3 of Table 2 in paper.


[1] Yang, Ling, et al. "DPGN: Distribution Propagation Graph Network for Few-shot Learning." Proceedings of the IEEE/CVF Conference on Computer Vision and Pattern Recognition. 2020
[2] Hu, Shell Xu, et al. "Empirical Bayes Transductive Meta-Learning with Synthetic Gradients." ICLR (2020)..

---

> ### Author Response · Authors · 2020-11-18
> **Reply**
>
> We want to thank the reviewer for your thorough reading and valuable comments! However, there are some points of misunderstanding that we address in this rebuttal.
>
> Q1. The writing is confusing and not clear enough. For example, Para. 4.1, line 6. What is the specific meaning of “main direction of n points”, and what is suitable mathematical expression of “the unit vector e” ?
>
> Thank you for pointing out these issues. We have corrected these issues and improved the writing in the new version.
>
> Q2. In Mini-Imagenet N-way K-shot experiments, authors didn’t show specific numbers of filter of the most important comparison object, Reptile, and the final experiments results are not particularly outstanding compared with recent papers, like DPGN[1], SIB[2].
>
> Reptile has 32 filters. In the new version, we have marked it in Table 1.
>
> Our focus is to improve the gradient-based meta-learner that is broadly applicable to other tasks in this work. And thus we do not compare some algorithms since they introduce additional information to improve accuracy. For instance,  [1] uses the pretrained models on 64 classes of entire meta-training set to complete feature extraction, which can significantly improve the algorithm's accuracy (earlier similar work such as [3,4,5]). The same problem exists in [2], so we did not compare [1,2]. But we will cite and discuss the difference between our work and [1,2] in Related Work.
>
> [3] Andrei A. Rusu, Dushyant Rao, Jakub Sygnowski, Oriol Vinyals, Razvan Pascanu, Simon Osin- dero, and Raia Hadsell. Meta-Learning with Latent Embedding Optimization. In International Conference on Learning Representations (ICLR), 2019.
>
> [4] Gidaris S, Komodakis N. Dynamic few-shot visual learning without forgetting[C]//Proceedings of the IEEE Conference on Computer Vision and Pattern Recognition. 2018: 4367-4375.
>
> [5] SiyuanQiao,ChenxiLiu,WeiShen,andAlanYuille.Few-ShotImageRecognitionbyPredicting Parameters from Activations. In IEEE Conference on Computer Vision and Pattern Recognition (CVPR), 2018.
>
> Q3. For label noise experiments, it is hard to say the ISPL is indeed effective as results showed in line 1,2,3 of Table 2 in paper.
>
> In Table 2, $p$ represents the proportion of noise. It can be found that as $p$ increases (0-0.5), the margin between ER and ER+ISPL becomes significantly larger. It means that the role of ISPL gradually becomes more important, and this rule also shows the robustness brought by ISPL. Besides, the experimental value is an average of five times. Even when $p$ is small, the numerical difference is significant.

---

### Official Review · AnonReviewer2 · 2020-10-30
**An interesting paper but has weakness on the theoretical part**

**Rating:** 5
**Confidence:** 4

**Review:**

This paper is concerned about the update of meta-parameters in gradient-based meta-learning approaches. Developed upon the recent method called Reptile (Nichol et al., 2018), this work aims to make it better deal with the sampling and label noises in few-short learning. The motivation is that Reptile updates meta-parameters towards the last task-specific parameters, but this could lead to biased updating. Instead, this work proposes to update the meta-parameters by the main direction of historical task-specific parameters to reduce the variation on gradients caused by the sampling and label noises. Theoretical analysis is conducted to show that the main direction of historical task-specific parameters is just the leading eigenvector of the covariance matrix computed upon the task-specific parameters obtained in the $n$ inner loop steps. This method is called Eigen-Reptile in this paper accordingly.

Further, this work proposes Introspective Self-paced Learning (ISPL) to improve the precision of the estimate of the main direction above. It creates multiple prior models by randomly sampling the training data and discards the samples with high loss values. Theoretical analysis is conducted to justify the proposed ISPL method.

Experimental study is conducted on a synthetic dataset and the Mini-ImageNet benchmark in few-shot classification tasks. The result shows that the proposed method can achieve better performance than Reptile and other state-of-the-art methods.

Overall, the issue researched in this paper has its significance, and the idea is sound. Also, experimental study demonstrates the effectiveness of the proposed method.

The main weaknesses of this work lie at its theoretical part, as detailed as follows.

1. Essentially, this work can be regarded as a kind of denoise method to better update the meta-parameters. Different from Reptile, this work proposes to update the meta-parameters by the "main direction" of task-specific parameters. Why is the "main direction" the best choice? The simplest choice could be the "mean" of the $n$ task-specific parameters. How is this option compared with the main direction? Please clarify.

2. The theoretical analysis in Section 4.1 is not novel. It is largely a derivation of principal component analysis which can be founded in textbooks. The only difference is that a linear constraint $\bar{V}e>0$ is imposed in Eq.(3). However, if understood correctly, the derivation seems to be flawed. This linear constraint does not have any effect to the optimal solution of Eq.(3), which is still proved as the leading eigenvector of $S$. Eq.(3) is a linear constrained quadratic programming. The leading eigenvector of $S$ does not necessarily satisfy the linear constraint related to an arbitrary $\bar{V}$. This may be related to some issue in the KKT condition listed in Eq.(5). Please double check or clarify it.

3. The trick in Eq.(6) is not new in machine learning and therefore cannot be regarded as a technical contribution. This trick has long been used to compute the eigensystems of the covariance matrix of "thin" or "fat" data matrices, for example, in the seminal work [R1].

4. The proof in Appendix B is largely routine. It mainly shows the properties of a covariance matrix when data are corrupted by white noise (that is, following a Gaussian distribution).

5. The proposed ISPL is interesting. However, the theoretical proof for Theorem 2 in the Appendix needs to be made clear. Particularly, the statements between Eq(29) and (30) are vague. Also, ISPL seems to be more related to ensemble learning. This paper may utilize existing ensemble learning theories to better justify this idea.

In short, this work is well motivated and the overall idea is sound. Experimental study supports the effectiveness of the proposed methods. Particularly, this paper makes efforts to provide theoretical justification for each of the methods, which is appreciated. However, these theoretical justifications currently are not strong or novel enough. Some of them need to be double checked and some need to be further clarified.

[R1] M. Turk; A. Pentland (1991). "Face recognition using eigenfaces" (PDF). Proc. IEEE Conference on Computer Vision and Pattern Recognition. pp. 586–591.

--- Thank the authors for the detailed response. After reading the response and the comments of peer reviewers, it is still felt that this work needs to better clarify some key issues and strengthen both theoretical and experimental study. In light of these, the rating is maintained as follows.

---

> ### Author Response · Authors · 2020-11-18
> **Reply(Part 1)**
>
> We want to thank the reviewer for your careful reading and providing a lot of critical comments! Below we address the concerns mentioned in the review:
>
> Q1. Essentially, this work can be regarded as a kind of denoise method to better update the meta-parameters. Different from Reptile, this work proposes to update the meta-parameters by the "main direction" of task-specific parameters. Why is the "main direction" the best choice? The simplest choice could be the "mean" of the n task-specific parameters. How is this option compared with the main direction? Please clarify.
>
> Yes, your insight is correct. However, we would like to highlight two points. First of all, averaging task-specific parameters cannot eliminate the effect of gradient noise like eigenvectors. Secondly, we conducted this experiment on the 5-way 1-shot task, and the result is 47.6 ± 0.17%, which is much lower than Eigen-Reptile with the same settings.
>
> Q2. The theoretical analysis in Section 4.1 is not novel. It is largely a derivation of principal component analysis which can be founded in textbooks. The only difference is that a linear constraint V¯e>0 is imposed in Eq.(3). However, if understood correctly, the derivation seems to be flawed. This linear constraint does not have any effect to the optimal solution of Eq.(3), which is still proved as the leading eigenvector of S. Eq.(3) is a linear constrained quadratic programming. The leading eigenvector of S does not necessarily satisfy the linear constraint related to an arbitrary V¯. This may be related to some issue in the KKT condition listed in Eq.(5). Please double check or clarify it.
>
> Thank you for your careful reading. However, we want to highlight that our main contribution is not in theoretical innovation but in applying past technologies to new scenarios and achieving good results. Our theoretical results only serve as the justification and guidance about the feasibility of the ER. We believe that the combination of casting the meta-overfitting issue as a gradient noise issue, introducing eigenvector into FSL to alleviate meta-overfitting on noise, and theoretically proving that the algorithm's effectiveness is novel.
>
> Eq(3) and Eq(5) are necessary for ER. They will determine whether the algorithm converges, as the required eigenvector may not satisfy the meta-learner update direction. We use Eq(3) in Appendix A Algorithm1 and algorithm implementation. When computing the eigenvector of S, Eq(3) and Eq(5) do not affect. They are written here because these formulas are necessary for implementing the algorithm. Simultaneously, the pseudo-code is in the appendix, so Eq(3) and Eq(5) are retained in this section to remind that the leading eigenvector needs to satisfy them.
>
> Q3.The trick in Eq.(6) is not new in machine learning and therefore cannot be regarded as a technical contribution. This trick has long been used to compute the eigensystems of the covariance matrix of "thin" or "fat" data matrices, for example, in the seminal work [R1].
>
> Thank you for pointing this out. We have removed this point in the contribution of the new version. But as we previously mentioned, our main contributions are to cast the meta-overfitting problem, to propose two algorithms based on different ideas, and to achieve state-of-the-art results. Our theories are provided to ensure the correctness of our algorithms.
>
> Q4.The proof in Appendix B is largely routine. It mainly shows the properties of a covariance matrix when data are corrupted by white noise (that is, following a Gaussian distribution).
>
> We mainly consider Gaussian noise because the gradient noise studied in many previous works [1,2,3] is also white noise.
>
> [1] Mandt S, Hoffman M, Blei D. A variational analysis of stochastic gradient algorithms[C]//International conference on machine learning. 2016: 354-363.
>
> [2] Jastrzębski S, Kenton Z, Arpit D, et al. Three factors influencing minima in sgd[J]. arXiv preprint arXiv:1711.04623, 2017.
>
> [3] Hu W, Li C J, Li L, et al. On the diffusion approximation of nonconvex stochastic gradient descent[J]. Annals of Mathematical Sciences and Applications, 2019, 4(1): 3-32.

---

> > ### Author Response · Authors · 2020-11-18
> > **Reply(Part2)**
> >
> > Q5.The proposed ISPL is interesting. However, the theoretical proof for Theorem 2 in the Appendix needs to be made clear. Particularly, the statements between Eq(29) and (30) are vague. Also, ISPL seems to be more related to ensemble learning. This paper may utilize existing ensemble learning theories to better justify this idea.
> >
> > Eq(29) indicates that the smaller the value of $\frac{\lambda_{o}}{\lambda}$, the more accurate the observed eigenvector. There are two situations for samples discarded by ISPL: 1. Classes that are not familiar to meta-learner. 2. Noisy data. This is because the impact of these two types of data on meta-learner is similar, and both are reflected in the larger loss value. So in Eq(30), it is assumed that their influence on the model parameters is $\xi$. Then, the numerator and denominator of $\frac{\lambda_{o}}{\lambda}$ are subtracted from $\xi$, the value of $\frac{\lambda_{o}}{\lambda}$ will become smaller, so the observed eigenvector will be more accurate.
> >
> > We highly appreciate your suggestions and we have revised our manuscript based on your comments in the new version.

---

### Official Review · AnonReviewer1 · 2020-11-01
**An novel and robust meta-learning algorithm**

**Rating:** 6
**Confidence:** 4

**Review:**

This paper proposes a novel and effective meta-learning algorithm using the so-called Eigen-Reptile to update the meta-parameters and address the gradient noises. To further improve the accuracy of the main direction of Eigen-Reptile, the authors employ self-paced learning algorithm to construct several prior models determining which samples should be abandoned. The experiments demonstrate strong robustness against label noises.

Strength:
- The proposed algorithm is technically sound. The authors provide the solution to address the high cost of computing eigenvalue and eigenvector in the ER process, and also offer the theorem to demonstrate that the eigenvectors will not be affected by gradient noises.
- The experiments are solid to demonstrate the usefulness of the Eigen-Reptile algorithm. Applying ER in regression, few-shot classification tasks achieve superior performance and robustness compared to the selected baseline models.

Weakness:
- Missing analyses on the training and testing overhead, e. .g, runtime, running memory.
- Missing comparisons to more recent meta-learning and few-shot learning approaches.
- ER is built upon a CONV-4 backbone. Is it applicable to more complicated architectures, e. g. ResNet-12 considering both the performance and overhead.

---

> ### Author Response · Authors · 2020-11-18
> **Reply**
>
> We are grateful to the reviewer for a nice summary, and for the kind recognition of our key contributions. Below we address the concerns mentioned in the review:
>
> Q1.Missing analyses on the training and testing overhead, e.g., runtime, running memory.
>
> We add the detailed complexity analysis of Eigen-Reptile in Appendix B of the new version, which shows that the spatial complexity is $\mathcal{O}(d)$ and time complexity is $\mathcal{O}(Td)$. $d$ is the number of network parameters, and $T$ is the maximal number of outer-loop. Simultaneously, Reptile has the same spatial and time complexity, which are much lower than the second-order optimization algorithms, $\mathcal{O}(Td^2)$.
>
> Q2. Missing comparisons to more recent meta-learning and few-shot learning approaches.
>
> We mainly focus on gradient-based meta-learning algorithms that do not introduce additional information to improve accuracy, so there are fewer existing works. However, we add some new baseline methods [1,2,3] into the table 1.
>
> [1]Yao H, Wu X, Tao Z, et al. Automated relational meta-learning[J]. arXiv preprint arXiv:2001.00745, 2020.
>
> [2] E. Park and J.B. Olivia, "Meta-curvature," Neurips, 2019.
>
> [3] Simon C, Koniusz P, Nock R, et al. On modulating the gradient for meta-learning[C]//European Conference on Computer Vision. Springer, Cham, 2020.
>
> Q3. ER is built upon a CONV-4 backbone. Is it applicable to more complicated architectures, e.g. ResNet-12 considering both the performance and overhead.
>
> Like Reptile, ER is model-agnostic, which means that it can combine any complex model. Our experiment is built upon the CONV-4 backbone for a fair comparison with many gradient-based meta-learning algorithms, as they all use the CONV-4 backbone. In addition, ER and Reptile have the same spatial and time complexity, which are much lower than the second-order meta-optimization algorithms. As second-order meta-optimization algorithms and Reptile are applicable to more complicated architectures, ER is undoubtedly applicable for a more complicated network, considering both the performance and overhead.

---

### Decision · Program_Chairs · 2021-01-07
**Final Decision**

**Decision:**

Reject

**Comment:**

This paper was evaluated by four reviewers. After rebuttal, several concerns remained, e.g. Rev. 1 is interested in more thorough comparisons even if the model is claimed to be backbone-agnostic. Rev. 2 is concerned about re-print of some theories and authors' response that 'contribution is not in theoretical innovation'. Rev. 3 is overall not impressed with the clarity of the paper. Finally, Rev. 4 also remains unconvinced after rebuttal due to several somewhat loose explanations provided by authors.

At this point, AC agrees with reviewers that the paper requires more clear-cut theoretical contributions, ablations and improvements in writing clarity. While some reviewers might have been more inspired by the aspect of noisy labels, even ignoring this aspect, the overall consensus among all reviewers stands.